# An interoperable web-based platform to support health surveillance against latent tuberculosis infection in health care workers and students: The evolution of CROSSWORD study protocol

Angela Rizzi[1,☯], Eleonora Nucera[1,2☯], Walter Mazzucco[3], Pierpaolo Palumbo[4], Domenico Staiti[5,6], Umberto Moscato[7,8], Francesco Maria De Simone[5], Michela Sali[9,10], Luca Boldrini[11], Nikola Dino Capocchiano[12], Stefano Patarnello[12], Gabriele Rumi[1], Raffaella Chini[1], Valentina Carusi[1], Michele Centrone[1], Alessia Di Rienzo[1], David Longhino[1], Chiara Laface[1], Sabato Mellone[13], Carmelo Massimo Maida[3], Emanuele Cannizzaro[3], Luigi Cirrincione[3], Maria Gabriella Verso[3], Annalisa Saracino[14], Francesco Di Gennaro[14], Luigi Vimercati[15], Luigi De Maria[15], Stefania Sponselli[15], Giancarlo Scoppettuolo[16], Roberta Pastorino[17], Antonio Gasbarrini[1,18,19], Riccardo Inchingolo[20]*

1 UOSD Allergologia e Immunologia Clinica, Dipartimento Scienze Mediche e Chirurgiche, Fondazione Policlinico Universitario A. Gemelli IRCCS, Rome, Italy, 2 Department of Translational Medicine and Surgery, Università Cattolica del Sacro Cuore, Rome, Italy, 3 Department of Health Promotion, Maternal and Infant Care, Internal Medicine and Medical Specialties (PROMISE), University of Palermo, Italy, 4 Department of Electrical, Electronic, and Information Engineering "Guglielmo Marconi" — DEI, University of Bologna, Italy, 5 UOSD Sorveglianza sanitaria, Fondazione Policlinico Universitario A. Gemelli IRCCS, Rome, Italy, 6 Dipartimento di Scienze della vita e sanità pubblica, Università Cattolica Sacro Cuore, Rome, Italy, 7 Section of Occupational Health, Health Surveillance and Radioprotection Health, University Department of Life Sciences and Public Health, Università Cattolica del Sacro Cuore, Rome, Italy, 8 UOC of Hospital Hygiene, Department of Women's, Children's and Public Health Sciences, Fondazione Policlinico Universitario Agostino Gemelli IRCCS, Rome, Italy, 9 Department of Laboratory and Infectious Sciences, Fondazione Policlinico Universitario A. Gemelli IRCCS, Rome, Italy, 10 Department of Basic Biotechnological Sciences, Intensive and Perioperative Clinics, Università Cattolica del Sacro Cuore, Rome, Italy, 11 UOC Radioterapia Oncologica, Fondazione Policlinico Universitario "A. Gemelli" IRCCS, Rome, Italy, 12 Gemelli Generator RWD, Fondazione Policlinico Universitario A. Gemelli IRCCS, Rome, Italy, 13 Health Sciences and Technologies-Interdepartmental Center for Industrial Research (HST-ICIR), University of Bologna, Bologna, Italy, 14 Dipartimento di Medicina di Precisione e Rigenerativa e Area Jonica (DiMePRe-J), Clinica Universitaria Malattie Infettive, Policlinico di Bari, Italy, 15 Interdisciplinary Department of Medicine (DIM), Unit of Occupational Medicine, University Hospital of Bari, 11 G. Cesare Square, , Bari, Italy, 16 UOC Malattie Infettive, Dipartimento di Scienze di Laboratorio e Infettivologiche, Fondazione Policlinico Universitario A. Gemelli IRCCS, Roma, Italy,, 17 Section of Hygiene, University Department of Life Sciences and Public Health, Università Cattolica del Sacro Cuore, Rome, Italy, 18 Department of Translational Medicine and Surgery, Università Cattolica del Sacro Cuore, Rome, Italy, 19 UOC Gastroenterologia and UOC CEMAD Medicina Interna e Gastroenterologia, Department of Medical and Surgical Sciences, Fondazione Policlinico Universitario Agostino Gemelli IRCCS, Rome, Italy, 20 UOC Pneumologia, Dipartimento Neuroscienze, Organi di Senso e Torace, Fondazione Policlinico Universitario A. Gemelli IRCCS, Rome, Italy

☯ These authors contributed equally to this work.
* riccardo.inchingolo@policlinicogemelli.it



**Data availability statement:** The minimal anonymized data set necessary to replicate study findings is openly available from BioStudies, S-BSST1682 (https://www.ebi.ac.uk/biostudies/studies/S-BSST1682. DOI: 10.6019/S-BSST1682).

**Funding:** This research was co-funded by the Italian Complementary National Plan PNC-I.1 "Research initiatives for innovative technologies and pathways in the health and welfare sector" D.D. 931 of 06/06/2022, "DARE - DigitAl life-long pRevEntion" initiative, code PNC0000002. The funders had no role in study design, data collection and analysis, decision to publish, or preparation of the manuscript.

**Competing interests:** The authors have declared that no competing interests exist.

# Abstract

## Background

Tuberculosis (TB) prevention is a major goal in teaching hospital setting. Because of the possible progression or reactivation of latent disease, the screening of both health-care workers (HCWs) and students is an important issue in the TB control program.

## Objective

to deploy a web-based platform interoperating health surveillance systems from different hospitals to define models based on the highlighted risk factors to predict the occurrence of Latent Tuberculosis Infection (LTBI) and to define prevention strategies and interventions.

## Methods

This is a cross-sectional ambispective observational study without drug and device. The primary endpoint is the prevalence of LTBI. The secondary endpoint is the identification of possible risk factors of LTBI in a large cohort of HCWs and students.

## Conclusions

This study aims to enrich the primary prevention measures against TB, having a high socio-economic-health impact in high-risk populations (HCWs and students) through an interoperable digital approach based on data obtained in three large Italian teaching hospitals.

**ClinicalTrials.gov:** NCT05756582

## Introduction

*Mycobacterium tuberculosis* (Mtb) may develop symptoms and signs of disease (TB disease), in a subset of subjects, or may have no clinical evidence of disease (latent tuberculosis infection [LTBI]) [1]. TB disease remains one of the major causes of morbidity and mortality in the world [1]. The WHO estimates that the annual incidence of tuberculosis has been raising worldwide from 10.0 million in 2020 to 10.6 million people in 2022 [1].

Mtb is transmitted from person to person via the airborne route [2]. Several factors determine the probability of Mtb transmission: (1) infectiousness of the source patient—a positive sputum smear for acid-fast bacilli (AFB) or a cavity on chest radiograph being strongly associated with infectiousness; (2) host susceptibility of the contact; (3) duration of exposure of the contact to the source patient; (4) the environment in which the exposure takes place (a small, poorly ventilated space providing the highest risk); and (5) infectiousness of the Mtb strain.

In 2021, a systematic review explored the status and risk factors of TB infection among HCWs through the evaluation of all published reports from 1 August 2010 to 31 December 2018. The TB infection rate in HCWs was higher than that of the general population. Based on 39 studies, the prevalence of TB in HCWs (tuberculin skin test positive) was 29.94%. In contrast, the global burden of latent TB infection was 23.0% (95% uncertainty interval:

20.4%–26.4%) in 2014. The risk factors of TB among HCWs were aging, long duration of employment, nursing professionals, lack of Bacillus Calmette-Guerin vaccination, and low body mass index [3].

In 2020, a meta-analysis examined population-level incident TB risk in a pooled data set of more than 80,000 individuals tested for LTBI in 20 countries with low M. tuberculosis transmission (annual incidence ≤ 20 per 100,000 persons). Cumulative 5-year risk of incident TB among people with untreated LTBI approached 16% among child contacts and approximately 5% among recent adult contacts, migrants from high TB-burden settings and immunocompromised individuals. Most cumulative 5-year risk was accrued during the first year among risk groups with an index exposure. The substantial variation in incidence rates even within these risk groups suggests that an individual-level approach to risk stratification is required. The first direct data-driven model was developed to incorporate the magnitude of the T cell response to M. tuberculosis with readily available clinical metadata to capture heterogeneity within risk groups and generate personalized risk predictions for incident TB [4].

Health-care workers (HCWs) are at higher risk of being exposed to tuberculosis compared to the general population [5], even in countries with low TB incidence, such as Italy. In these workers, the assessment of LTBI and TB disease is part of the annual health surveillance program aimed to prevent occupational diseases [6].

Because of the severity of their symptoms, individuals with an active contagious form of tuberculosis have a higher chance of being hospitalized and can remain for hours or days at the health care facility with a pending diagnosis.

For these reasons, the prevention of nosocomial transmission of tuberculosis represents prominent public health and occupational requirement [7,8]. However, it is also important to note that health-care providers (HCPs) may contract an infectious disease outside of the workplace, thus representing a source of infection to colleagues and patients that are more likely to be affected.

It is estimated that one patient with untreated active infectious pulmonary tuberculosis can affect 10–15 people within a year [9]. Of these individuals, a percentage between 3 and 10% will be infected, half of whom will become ill within one to two years while the others might develop the illness at any point of their residual life [7]. On average, 10% of people with LTBI develop TB disease in their lifetime, half of which become ill within 2 years after exposure [5,7,9]. About 70% of incident cases of TB are a result of the reactivation of a past infection [5,7,9,10], especially in countries with a low TB prevalence [11–13]. Furthermore, the pharmacological management of patients with TB can be complicated by drug-induced reactions [14].

For these reasons, the identification and prophylactic treatment of individuals with LTBI are crucial for the elimination of the disease. Health-care students involved in clinical training could be exposed to occupational risks like those of HCWs. Therefore, the screening for LTBI of both HCWs and undergraduate students attending teaching hospitals is recommended especially in low-incidence countries, including Italy, in order to obtain an early diagnosis of cases and prevent progression to active disease [8,15].

Despite the higher risk of TB infection reported in high TB burden countries among both HCWs and healthcare students compared to the general population [16,17], such TB control activities are difficult to be implemented in these settings [18].

According to Italian law [19,20], it is mandatory for employers to evaluate all risks in the workplace, including the risk of exposure to biological agents, and to implement measures to prevent the spread to operators or to limit the severity of its consequences. For this purpose, great importance is given to health surveillance of tuberculosis infection. In fact, the surveillance of LTBI in HCWs is considered fundamental for tuberculosis prevention.

Very few studies have investigated the epidemiology of TB infection among undergraduate healthcare students worldwide, and few, with a large sample, in areas with a low-incidence of TB [17,21–27].

The ongoing CROSSWORD study, aimed at collecting big data on the prevalence of LTBI in HCWs and students and the identification of possible risk factors for LTBI, in a single center, presents, as main current criticality, the monocentric nature of the study. The translation of the same methodologies of the CROSSWORD study within the research project entitled "*Implementing an interoperable web-based platform to support health surveillance against latent tuberculosis infection in health care workers and students to define prevention strategies and interventions: the evolution of CROSSWORD Study*" and the involvement of more centers will allow the creation of a shared interoperable digital platform and the definition of primary prevention strategies for LTBI.

Our research project is inspired by TB risk prediction model elaborated in a recent meta-analysis (PERISKOPE-TB) that combines a quantitative measure of T cell sensitization and clinical covariates [4]. We adopted some candidate predictors included in the final version of prognostic model described by Gupta et al.: exposure, HIV status and history of hematological or solid organ transplant [4].

Here we present the protocol of the Evolution of CROSSWORD study.

## Materials and methods

### The aim

The Evolution of CROSSWORD Study is an ambispective observational study aimed at: 1) estimating the prevalence of LTBI in HCWs and students attending University Hospitals with focus on the association between LTBI and potential risk factors, and 2) creating a web-based platform interoperating health surveillance systems from different hospitals to define models based on the highlighted risk factors to predict the occurrence of LTBI and to define prevention strategies and interventions.

### Design

The Evolution of CROSSWORD study includes a retrospective phase focused on the analysis of data derived from the ongoing CROSSWORD Study conducted at Fondazione Policlinico Universitario A. Gemelli IRCCS (FPG) and Università Cattolica del Sacro Cuore (UCSC) in Rome, Italy. The subsequent prospective phase will collect data from two centers in southern Italy, the University of Palermo (UNIPA) and the University of Bari (UNIBA)/ University Hospital of Bari. It is part of the national research project "DARE - Digital Lifelong Prevention", funded by the Ministry of University and Research (MUR) as part of the National Plan for Complementary Investments to the PNRR, with the goal of maximizing the potential of data to enhance health promotion and enable lifelong prevention. DARE focuses on improving health trajectories by leveraging data. Furthermore, DARE Project aims to create a community focused on digital prevention through research, innovation, and engagement of all stakeholders [28].

With the support of the University of Bologna, partner of the national DARE project, the Research Team will then proceed with the analysis of the data collected at three clinical centers and the definition of a single data platform necessary for the assessment of the prevalence of LTBI and the identification of risk factors. Furthermore, prediction algorithms will be validated in all university hospitals involved through the recruitment of a new cohort of HCWs and students.

## Setting of the study

The participants will be recruited at: 1) the Health Surveillance Unit of Fondazione Policlinico Universitario A. Gemelli IRCCS, in Rome (FPG and UCSC), 2) the Department of Health Promotion, Maternal and Infant Care, Internal Medicine and Medical Specialties (PROMISE) of University of Palermo (UNIPA), 3) Dipartimento di Medicina di Precisione e Rigenerativa e Area Jonica (DiMePRe-J) of University of Bari (UNIBA) and 4) Unit of Occupational Medicine of University Hospital of Bari.

## The sample size

From the literature, Coppeta et al. [29] found a rate of LTBI prevalence of 2.1% in a population of Italian HCWs in Rome, and Verso et al. reported a prevalence rate of LTBI of 0.62% among Italian healthcare students and postgraduates in Palermo [26]. Therefore, we assumed a prevalence of LTBI of 1% in our population. The sample size was calculated considering the objectives of estimating the prevalence of LTBI and identifying its risk factors.

As regards the estimate based on the first outcome, considering 1) the assumed prevalence of LTBI of P = 1% in each teaching hospital involved in the research project, 2) an error estimate of d = 0.3%, and 3) a chosen confidence level of 90%, we obtained a total sample size of 2,977 subjects to be enrolled. The following formula [30] was used:

$$n = \frac{Z^2 \; P \; (1 - P)}{d^2}$$

where Z = Z statistic for a level of confidence (1.64 for 90% confidence level) [30]

The potential involvement of all employees and students attending medical, specialization and health professions degree courses, in the three university hospital centers involved in the project, will reasonably allow us to reach a higher sample size than the minimum target of 2,977 subjects. Indeed, for detecting an odds ratio (OR) as low as 1.69 (OR for male gender in [29]) with a type I error $\alpha$=0.05 and power $\beta$=0.95, it would be required to reach a sample size of 3,503 subjects (z-test on logistic regression)

The calculation was performed using R statistical software (R Core Team, 2021).

## Participants

During the study period, all HCWs and students, who refer to Health Surveillance Unit of FPG/UCSC, UNIPA and to Unit of Occupational Medicine of University Hospital of Bari for the annual scheduled medical visit for judgment of suitability for work/activity, will be invited to participate to the study giving written informed consent. Participation in the study is proposed to all HCWs and students of three University Hospitals involved in the project (Fig 1). There are no other specific inclusion criteria. Exclusion criteria include known history of tuberculosis, previous chemoprophylaxis, positive tuberculin skin test (TST) performed within 12 months prior to enrollment and refusal of informed consent.

## Procedures

### Primary outcome

The general objective of the research project is to develop an interoperable web-based platform to provide epidemiological and health surveillance against LTBI in the hospital setting. The primary outcome of the study is the prevalence of LTBI defined as 1) positive TST at enrollment, 2) tuberculin conversion defined as negative TST before enrollment and positive at enrollment, 3) subsequent positive QuantiFERON-TB Gold (QFT) test result performed on the same day as

| | Enrolment | Allocation | Post-allocation | Close-out |
|---|---|---|---|---|
| **TIMEPOINT** | $-t_1$ | $0$ | $t_1$ | $t_2$ |
| **ENROLMENT:** | | | | |
| **Eligibility screen** | X | | | |
| **Informed consent** | X | | | |
| **Allocation** | | X | | |
| **INTERVENTIONS:** | | | | |
| *[Questionnaire]* | | | ←——→ | |
| **ASSESSMENTS:** | | | | |
| *[Baseline variables]* | X | X | | |
| *[Prevalence and risk factors]* | | | | X |

**Fig 1. Schedule of enrolment, interventions, and assessments.**

the positive TST result or within 5 days of tubulin conversion [higher response than the cut-off value of 0.35 IU/ ml of INF-γ was detected in at least one test tube (TB1 or TB2)], 4) chest imaging excluding pleural and/or pulmonary lesions suggestive of active tuberculosis and 5) final medical evaluation done by a Specialist in Infectious Diseases which excluded active disease. All five criteria must be met. One TST is scheduled for each participant. Furthermore, the study does not provide follow-up. Each participant continues regular annual follow-up as per routine clinical practice.

## Secondary outcomes

The secondary outcomes are a) the identification of possible risk factors of LTBI in a large cohort of HCWs and students, and b) the validation of the prediction models through the enrolment of a subsequent new cohort of HCWs and students at the teaching hospitals.

The following list of possible risk factors will be recorded:

- Age and sex.

- Bacillus Calmette–Guérin (BCG) vaccination history.

- Previous work or stay in other countries with high TB rates including Armenia, Azerbaijan, Belarus, Bulgaria, Estonia, Georgia, Kazakhstan, Kyrgyzstan, Latvia, Lithuania, Moldova, Romania, Russia, Tajikistan, Turkey, Turkmenistan, Ukraine, and Uzbekistan.

- Known recent (within 6 months) exposure to TB-infected patient, which makes HCW or student a close contact of individuals with tuberculosis [31].

- Current smoking.

- Co-morbidities/ medical history (HIV, diabetes, silicosis, chronic kidney disease, solid organ transplant, malignant hematologic malignancies or undergoing chemotherapy, gastrectomy or fasting bypass, biological drugs).

- Median QuantiFERON-TB Gold In-Tube level (TB Ag-Nil).

- Occupation (doctor, nurse, laboratory technicians, paramedical personnel, "preclinical" students with no contact with patients inside the hospital, and "clinical" students attending various medical, laboratory, and surgical departments of the hospital, including the infectious disease wards).

- Working/ training Unit

- Years working as HCW

## Data management plans

All enrolled participants, who will agree to participate in the study and answer the questionnaire, will respond to a cross-sectional questionnaire survey through dedicated tool only once.

The survey contains a brief explanation of the study aims and an invitation to respond to a 9-item multiple-choice questionnaire.

During the first phase of the project, the Research Team adopted Microsoft Forms included in Microsoft Office 365 subscription of FPG and UCSC. Once the results start to roll in, the survey author is then able to use the simple built – in analytic tools within Forms to analyze the results and construct them into reports such as Excel tables. Although there are other surveying tools available (Google Forms, Survey Monkey etc.) Microsoft Forms is the standard for FPG and UCSC as all data is held securely within the university's Office 365 tenancy. The questionnaire is created following instructions available in the Microsoft Forms web site, and responses are automatically re-directed to the Research Team.

A specific 9 items self-test questionnaire, developed for the specific purpose and never administered before, is administered to the enrolled HCWs and students (Fig 2).

The questionnaire was firstly validated through the following procedure

- A Delphi Method structured on a panel of experts to achieve mutual agreement.

- The questionnaire was the object of a first validation phase on the first 25 responding subjects and results were analyzed to test its applicability and generalizability.

In addition to demographic information, a list of possible risk factors includes age and sex, Bacillus Calmette–Guérin (BCG) vaccination history, previous work or stay in other countries with high TB rates, known recent (within 6 months) exposure to TB-infected patient, current smoking, co morbidities/ medical history, median QuantiFERON-TB Gold In-Tube level (TB Ag-Nil) if available, occupation, working/ training Unit and years working as HCWs [32]. All positive QFT results on the first determination are repeated to confirm the finding. All conversions of QFTs are collected for the final analysis. No incentives will be offered to complete the survey. A first predictive model for LTBI will be set up through analysis performed by Real World Data GENERATOR.

All HCWs and students enrolled in UNIPA and in University Hospital of Bari will undergo the same methodologies adopted in FPG/UCSC.

1. What is your country of birth?

[ ] Italy
[ ] UE
[ ] Non UE

2. Have you ever had anti-tubercular vaccination?

[ ] Yes
[ ] No
[ ] I do not know/I do not remember

3. Are you a current smoker?

[ ] Yes
[ ] No
[ ] I prefer not to answer

4. Have you worked or stayed, for at least 3 months, in other countries with high TB rates? (High priority countries - Hpc) that include Armenia, Azerbaijan, Belarus, Bulgaria, Estonia, Georgia, Kazakhstan, Kyrgyzstan, Latvia, Lithuania, Moldova, Romania, Russia, Tajikistan, Turkey, Turkmenistan, Ukraine and Uzbekistan.

[ ] Yes
[ ] No

5. Have you had recent (within 6 months) exposures to TB-infected patients?

[ ] Yes
[ ] No
[ ] I do not know/I do not remember

6. Do you have one of the following clinical conditions: HIV, diabetes, silicosis, chronic kidney disease, solid organ transplant, malignant hematologic malignancies?

[ ] Yes
[ ] No
[ ] I prefer not to answer

7. Are you undergoing chemotherapy?

[ ] Yes
[ ] No
[ ] I prefer not to answer

8. Have you had gastrectomy or fasting bypass?

[ ] Yes
[ ] No
[ ] I prefer not to answer

9. Are you taking alpha tumor necrosis factor therapy (TNF-alpha) or other biological agents?

[ ] Yes
[ ] No
[ ] I prefer not to answer

**Fig 2. Health care workers and students questionnaire (English version).** The specific 9 items self-test questionnaire, developed for the specific purpose and never administered before, administered to the enrolled HCWs and students.

## Statistical analysis

Descriptive statistics will be performed to characterize responses to the survey. The distribution of the qualitative variables will be shown in percentages, whereas the quantitative variable means, and standard deviations will be used after performing the Shapiro-Wilk test to verify

their parametric distribution. Categorical variables will be compared using the Chi-squared test.

Logistic regression analysis will be performed to assess the association between LTBI and possible risk factors including age and sex, Bacillus Calmette–Guérin (BCG) vaccination history, previous work or stay in other countries with high TB rates, known recent (within 6 months) exposure to TB-infected patient, current smoking, co morbidities/ medical history, median QuantiFERON-TB Gold In-Tube level (TB Ag-Nil) if available, occupation, working/ training Unit and years working as HCWs.

The results of the risk-assessment analysis will be stratified according to sex, nationality, TB incidence of the country of birth and BCG vaccination and possible change in the roles of the participants and consequently in the risks of exposure.

Specific data visualization techniques will be used to provide a qualitative representation of the outcomes.

For the purposes of the project, cloud-based applications will be used to collect and store the data coming from the three healthcare centres. Data analytics will be adopted to measure the associations of risk factors with LTBI development. Finally, machine learning techniques will be used to predict the presence of LTBI with respect to health data and identified risk factors.

The final deliverable of the project will be a computerized decision support system able to deliver alerts in the case of LTBI presence.

*p-values* ≤ 0.05 will be considered statistically significant.
All the statistical analyses will be carried out using R statistical software (R Core Team, 2021).

## Ethical considerations and declarations

The authors declare that the procedures will follow the ethical standards of the Declaration of Helsinki and the guidelines of good clinical practice. The Territorial Ethics Committee (CET) Lazio Area 3 has given its approval for the amendment prot. 3 of October 2nd, 2023 (ID: 3528, Prot. N. 0000627/23, December 11th, 2023), relating to the CROSSWORD Study (ID: 3528, Prot. N. 0046362/20, November 16th, 2020).

The authors confirm that all ongoing and related trials for this intervention are registered on ClinicalTrials.gov [NCT05756582].

The document informing participants about the study contains the reasons for completing the questionnaire and the potential benefits arising from analyzing the results. The participants will be informed that participation in the study is completely voluntary and that they can withdraw from the study at any time without negative consequences. The answers to the questionnaires will be treated confidentially and anonymously, and the handling of the data will also be confidential. The members of the Research Team will hand out the consent forms to the participants and will be available to answer any questions they may have about the study. The site PI and Co-PIs had access to information that could identify individual participants during or after data collection.

## The status and timeline of the study

The retrospective phase of the project started on November 17, 2021. The final data analysis will be completed by November 2024. The multicenter prospective phase started in September 2024. The estimated primary completion date is June 30, 2026. The estimated study completion date is December 15, 2026. The expected time for publication of results is the last quarter of 2026 (Fig 3).

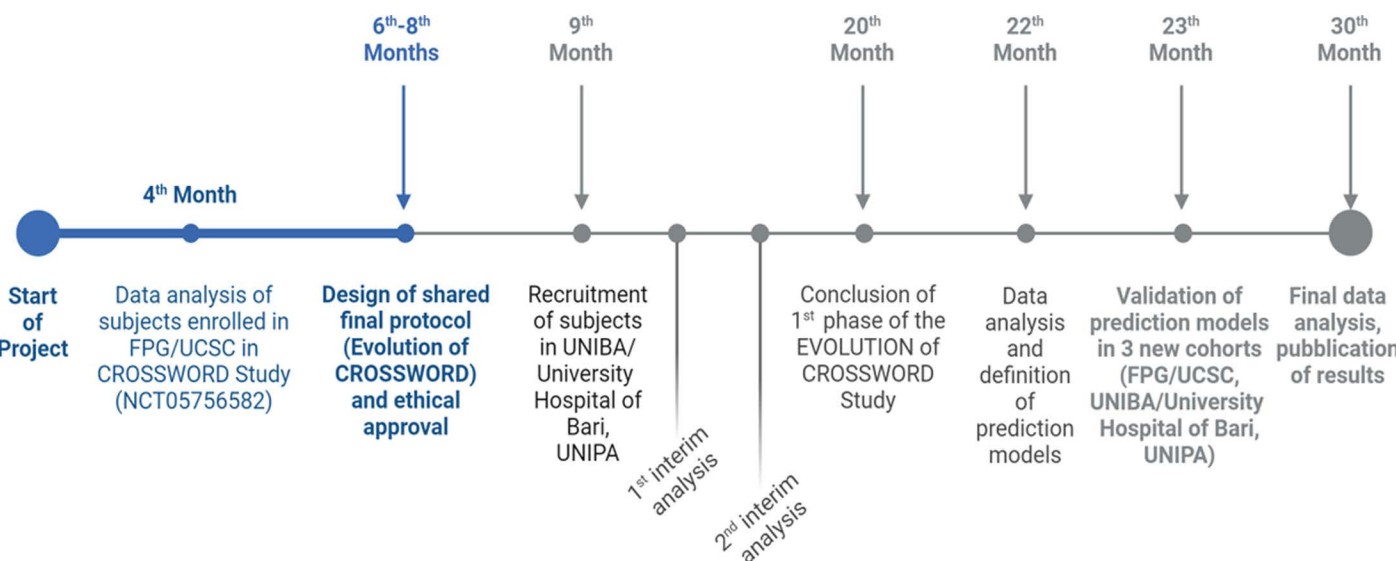

**Fig 3. Chrono program of the Evolution of CROSSWORD Study.** The Evolution of CROSSWORD study is an ambispective study including a retrospective phase focused on the analysis of data derived from the ongoing CROSSWORD Study, conducted at Fondazione Policlinico Universitario A. Gemelli IRCCS (FPG) and Università Cattolica del Sacro Cuore (UCSC) in Rome, and a prospective phase during which data will be collected from the University of Palermo (UNIPA) and the University of Bari (UNIBA)/ University Hospital of Bari, Italy.

## Discussion

This study aims to enrich the primary prevention measures against TB, having a high socio-economic-health impact in high-risk populations (health-care workers and students) through an interoperable digital approach based on data obtained in three large Italian teaching hospitals (Fondazione Policlinico Universitario A. Gemelli IRCCS and Università Cattolica del Sacro Cuore, University of Palermo, and University Hospital of Bari).

The purpose of conducting LTBI prevention program for HCWs and students is to block the spread of TB by preventing infected persons from progressing to active TB. However, currently available diagnostic tests for LTBI have several limitations, such as variable specificity.

The enrolment of many HCWs and students through an ambispective clinical observation design will provide robust data on the prevalence of LTBI. Furthermore, the study will allow us to identify risk factors associated with the occurrence of LTBI. This data will make it possible to develop an interoperable digital platform integrating the health surveillance systems already in use with other data flows (medical record). Next, prediction models of LTBI, based on the identified risk factors, will be outlined. Finally, the accuracy of the prediction model will be evaluated through a validation phase which will include a new cohort of HCWs and students enrolled in the aforementioned teaching hospitals.

After initiation of the trial, any changes to the protocol that impact the study objectives, study design, subject population, study procedures, or significant management aspects will require a formal protocol amendment. Any changes must be approved by Ethics Committees before implementation. Any deviation from the protocol must be documented and justified by a rationale and be subjected to approval. Any deviation from the protocol will be documented and justified by a rationale and will be subjected to approval. There will be no request for additional consent for the collection and use of data and biological materials from participants in ancillary studies. All complete confidential information relating to the study will be managed by the core Research Team (PI, Co-PI e site PI). Furthermore, the core Research Team will be

responsible for the quality of data collection and their confidentiality. Only the team involved in the study will have access to the study documents which will be kept in a secure environment in accordance with ICH-GCP. Patient personal data, which is included in the investigator database, will be processed in accordance with all applicable laws and regulations. To protect patient identity, a unique patient identification code will be assigned to each patient/subject in the study and will be used in place of the patient/subject's name when the investigator reports any study data. Therefore, this number, rather than the patient/subject's name, will appear on all documents and will be referenced to the patient/subject's date of birth.

The researchers intend to communicate the results of the trial to participants, healthcare professionals, and other relevant groups through scientific publications and national and international conference proceedings.

## Author contributions

**Conceptualization:** Angela Rizzi, Eleonora Nucera, Domenico Staiti, Gabriele Rumi, Riccardo Inchingolo.

**Data curation:** Angela Rizzi, Pierpaolo Palumbo, Raffaella Chini.

**Formal analysis:** Pierpaolo Palumbo, Nikola Dino Capocchiano, Stefano Patarnello, Sabato Mellone.

**Investigation:** Angela Rizzi, Umberto Moscato, Francesco Maria De Simone, Michela Sali, Valentina Carusi, Michele Centrone, Alessia Di Rienzo, David Longhino, Chiara Laface, Carmelo Massimo Maida, Emanuele Cannizzaro, Luigi Cirrincione, Maria Gabriella Verso, Annalisa Saracino, Francesco Di Gennaro, Luigi Vimercati, Luigi De Maria, Stefania Sponselli, Giancarlo Scoppettuolo.

**Methodology:** Angela Rizzi, Pierpaolo Palumbo, Luca Boldrini, Raffaella Chini, Riccardo Inchingolo.

**Supervision:** Eleonora Nucera, Walter Mazzucco, Roberta Pastorino, Antonio Gasbarrini.

**Writing – original draft:** Angela Rizzi, Pierpaolo Palumbo, Riccardo Inchingolo.

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
