## [Decision Letter · Decision Letter 0]

29 Sep 2024

PONE-D-24-34897Protocol the Evolution of CROSSWORD Study: Implementing an interoperable web-based platform to support health surveillance against latent tuberculosis infection in health care workers and students to define prevention strategies and interventionsPLOS ONE

Dear Dr. Inchingolo,

Thank you for submitting your manuscript to PLOS ONE. After careful consideration, we feel that it has merit but does not fully meet PLOS ONE’s publication criteria as it currently stands. Therefore, we invite you to submit a revised version of the manuscript that addresses the points raised during the review process.

**ACADEMIC EDITOR:**

Dear Authors, 

  the manuscript is interesting, but it needs some changes. Both reviewers suggested some changes to be made that I think are appropriate. Finally, I suggest providing more details in the sample size section, describing how the percentages were identified, and the sample size formula used.

Best regards,

Nicola, Serra

We look forward to receiving your revised manuscript.

Kind regards,

Nicola Serra

Academic Editor

PLOS ONE

Journal Requirements:

2. We note that you have selected “Clinical Trial” as your article type. PLOS ONE requires that all clinical trials are registered in an appropriate registry (the WHO list of approved registries is at      https://www.who.int/clinical-trials-registry-platform/network/primary-registries" https://www.who.int/clinical-trials-registry-platform/network/primary-registries and more information on trial registration is at http://www.icmje.org/about-icmje/faqs/clinical-trials-registration/ ).  

Please state the name of the registry and the registration number (e.g. ISRCTN or ClinicalTrials.gov) in the submission data and on the title page of your manuscript.

    a) Please provide the complete date range for participant recruitment and follow-up in the methods section of your manuscript.

    b) If you have not yet registered your trial in an appropriate registry, we now require you to do so and will need confirmation of the trial registry number before we can pass your paper to the next stage of review. Please include in the Methods section of your paper your reasons for not registering this study before enrolment of participants started. Please confirm that all related trials are registered by stating: “The authors confirm that all ongoing and related trials for this drug/intervention are registered”.

Please see http://journals.plos.org/plosone/s/submission-guidelines#loc-clinical-trials for our policies on clinical trials.

“This research was co-funded by the Italian Complementary National Plan PNC-I.1 "Research initiatives for innovative technologies and pathways in the health and welfare sector” D.D. 931 of 06/06/2022, "DARE - DigitAl lifelong pRevEntion" initiative, code PNC0000002.”

4. We note that you have indicated that there are restrictions to data sharing for this study. PLOS only allows data to be available upon request if there are legal or ethical restrictions on sharing data publicly. For more information on unacceptable data access restrictions, please see http://journals.plos.org/plosone/s/data-availability#loc-unacceptable-data-access-restrictions .  

5. Please include a caption for figures 2 and 3.

6. Please include captions for your Supporting Information files at the end of your manuscript, and update any in-text citations to match accordingly. Please see our Supporting Information guidelines for more information: http://journals.plos.org/plosone/s/supporting-information .

Reviewers' comments:

Reviewer's Responses to Questions

**Comments to the Author**

1. Does the manuscript provide a valid rationale for the proposed study, with clearly identified and justified research questions?

Reviewer #1: Yes

Reviewer #2: Yes

2. Is the protocol technically sound and planned in a manner that will lead to a meaningful outcome and allow testing the stated hypotheses?

Reviewer #1: Yes

Reviewer #2: Partly

3. Is the methodology feasible and described in sufficient detail to allow the work to be replicable?

Reviewer #1: No

Reviewer #2: Yes

4. Have the authors described where all data underlying the findings will be made available when the study is complete?

Reviewer #1: No

Reviewer #2: No

5. Is the manuscript presented in an intelligible fashion and written in standard English?

Reviewer #1: Yes

Reviewer #2: No

6. Review Comments to the Author

You may also provide optional suggestions and comments to authors that they might find helpful in planning their study.

Reviewer #1: The protocol titled "Protocol the Evolution of CROSSWORD Study: Implementing an interoperable web-based platform to support health surveillance against latent tuberculosis infection in healthcare workers and students to define prevention strategies and interventions" is well-structured and addresses a crucial area in tuberculosis (TB) prevention. This study has the potential to make a significant impact on the development of TB preventive strategies in low-TB-burden countries, especially for healthcare workers (HCWs) and students. However, some aspects need clarification and further elaboration:

1. Methods for LTBI Detection Tests:

o The manuscript did not clearly define the methods for latent tuberculosis infection (LTBI) detection tests. Both Interferon-Gamma Release Assay (IGRA) and Tuberculin Skin Test (TST) were mentioned in determining the primary endpoint. It is essential to specify whether the IGRA or TST will be used. Since the study involves the conversion of TST, multiple tests are assumed to be done. Information on how often follow-up testing will occur is needed to provide a comprehensive understanding of the testing process.

2. Interaction with the Developed Web-Based Framework:

o The article mentions the development of a web-based interactive platform but does not elaborate on how this framework will engage in the prediction and detection of LTBI. It would be helpful to describe how the platform will interact with collected data, monitor participants, or offer decision support for LTBI prediction.

3. Rationale for the Estimated Prevalence:

o The protocol estimates an LTBI prevalence of 1% in the participating institutes. However, the rationale behind this estimate is not sufficiently explained. Is it derived from the ongoing CROSSWORD study? Clarifying how this estimate was derived, considering factors such as the previous prevalence rates in similar settings or literature, would strengthen the study's methodology.

4. Consideration of Changes in Participants’ Roles Over Time:

o Since the study involves a potentially prolonged follow-up period, there is a high likelihood that participants’ roles and exposure risks may change (e.g., from students to practicing HCWs). This factor should be taken into account when identifying risk factors and analyzing LTBI risk. Clarifying how these changes will be tracked and integrated into the analysis is crucial for ensuring accurate risk assessment.

Overall Recommendations:

• The study protocol is valuable and has the potential to contribute significantly to TB prevention efforts. Addressing these points will enhance the clarity and comprehensiveness of the study. The authors are encouraged to provide detailed explanations regarding the LTBI detection protocols, web-based framework interaction, rationale for prevalence estimation, and adjustments for role changes during the study follow-up.

Reviewer #2: The authors describe the protocol of a longitudinal study (retrospective and prospective), aimed at providing a large amount of data on the incidence and prevalence of latent tuberculosis infection, on the incidence of tuberculosis and on the results of screening tests among health workers and students enrolled in 3 university hospitals in Central and Southern Italy. The objective would be achieved through the digital registration of some selected variables reported in a self-administered questionnaire. The objective is ambitious and the study protocol could validly contribute to scientific progress on a pathology present throughout the world, slightly increasing after the pandemic and with a significant socioeconomic burden even in low-incidence countries, such as Italy. Moreover, university students, an undestudied population strata, are included.

However, I think that the submitted description of the protocol should be enriched with details and few methodological aspects should be further clarify. I recommend substantial revisions that take into account the considerations reported below.

First of all, it's necessary to know how many employees and students are in the 3 hospitals considered, this to have information on the feasibility of reaching the sample size in the expected period. It might be useful to know how many people are divided into different age groups, how many students, whether only medical courses or other types of courses, such as engineering, are included, how many men and how many women. It might also be useful to know what is the incidence of tuberculosis in the reference population of the 3 hospitals considered and what is the estimated incidence of active or latent tuberculosis in previous periods among the healthcare workers and students. As regards the feasibility, add information on the progress of the Crossword study, which represents the retrospective phase of the study protocol. Moreover, it's necessary to provide more details on which aspects of the Periskope-TB study were captured in the described protocol, to better support the choice of selected variables.

The DARE project is mentioned but its contribution to this protocol is unclear. More information on the Dare study (blibliographic citation, if available) could clarify how the predictive algorithms adopted in this protocol were digitized and created.

It's necessary to stratify the sample size calculation by workers and students, sex and age groups, considering that the enrolled population includes workers and students probably in a wide age range.

Since workers and students are subjected to annual health surveillance, it is not clear whether they will answer only once or multiple times (and how many?) to the proposed questionnaire.

Authors should consider adding to the exclusion criteria, having had tuberculosis or having already undergone chemoprophylaxis .

About the primary outcomes, consider to add a symptom questionnaire. It should also be clarified how to consider the possible movement between working units.

Among the secondary outcomes, it would be better to add a definition of exposure to TB-infected patients, to avoid different descriptions

I would like the authors to explain the reasons why they collect health information through a self-administered questionnaire and not through medical records, which would avoid recall bias and inaccuracies. For question 4 I would put an indicative time (1 month or 3 months of stay).

Consider to add a table or a figure to list the recorded variables . Line 243: it's not clear if the items listed will be recorded.

Furthermore, authors should reread the text to make the English grammar more fluent. Few typos should be corrected: uniform the acronym TB or TBC, replace the word tuberculosis with the somewhat long acronym in the text; add title and description of figure 2 and 3.

I would encourage the authors to consider making the dataset public, possibly removing variables that could identify the respondent (e.g. department, course of study, etc.).

Authors should check bibliography according to the journal's instructions, in particular citations 1, 5, 6, 7, 9, 19, 20.

7. PLOS authors have the option to publish the peer review history of their article (what does this mean? ). If published, this will include your full peer review and any attached files.

**Do you want your identity to be public for this peer review?** For information about this choice, including consent withdrawal, please see our Privacy Policy .

Reviewer #1: No

Reviewer #2: No

---

## [Author Response · Author response to Decision Letter 0]

18 Oct 2024

October, 10th 2024

To Editor and Reviewers

PLOS ONE

We would like to greatly thank the Editor and Reviewers who encouraged a revision of the manuscript.

Please find enclosed the revised version (vers. 2) of the Study Protocol article entitled “Protocol the Evolution of CROSSWORD Study: Implementing an interoperable web-based platform to support health surveillance against latent tuberculosis infection in health care workers and students to define prevention strategies and interventions” by Angela Rizzi, Eleonora Nucera, Walter Mazzucco, Pierpaolo Palumbo, Domenico Staiti, Umberto Moscato, Francesco Maria De Simone, Michela Sali, Luca Boldrini, Nikola Dino Capocchiano, Stefano Patarnello, Gabriele Rumi, Raffaella Chini, Valentina Carusi, Michele Centrone, Alessia Di Rienzo, David Longhino, Chiara Laface, Sabato Mellone, Carmelo Massimo Maida, Emanuele Cannizzaro, Luigi Cirrincione, Maria Gabriella Verso, Annalisa Saracino, Francesco Di Gennaro, Luigi Vimercati, Luigi De Maria, Stefania Sponselli, Giancarlo Scoppettuolo, Roberta Pastorino, Antonio Gasbarrini and Riccardo Inchingolo.

PONE-D-24-34897

Protocol the Evolution of CROSSWORD Study: Implementing an interoperable web-based platform to support health surveillance against latent tuberculosis infection in health care workers and students to define prevention strategies and interventions

PLOS ONE

ACADEMIC EDITOR:

Dear Authors,

the manuscript is interesting, but it needs some changes. Both reviewers suggested some changes to be made that I think are appropriate. Finally, I suggest providing more details in the sample size section, describing how the percentages were identified, and the sample size formula used.

We thank the Academic Editor for the comment. We reviewed and reported the formula adopted for the sample size based on both the first and the second outcomes. We modified the manuscript accordingly. Lines: 222-240.

Journal Requirements:

We thank for the comment. The manuscript was modified accordingly.

2. We note that you have selected “Clinical Trial” as your article type. PLOS ONE requires that all clinical trials are registered in an appropriate registry. Please state the name of the registry and the registration number (e.g. ISRCTN or ClinicalTrials.gov) in the submission data and on the title page of your manuscript.

We thank for the comment. The manuscript was modified accordingly. Lines: 89-90 and 348-349.

a) Please provide the complete date range for participant recruitment and follow-up in the methods section of your manuscript.

We thank for the comment. The manuscript was modified accordingly. Lines: 361-364.

b) If you have not yet registered your trial in an appropriate registry, we now require you to do so and will need confirmation of the trial registry number before we can pass your paper to the next stage of review. Please include in the Methods section of your paper your reasons for not registering this study before enrolment of participants started. Please confirm that all related trials are registered by stating: “The authors confirm that all ongoing and related trials for this drug/intervention are registered”.

We thank for the comment. The trial was registered on ClinicalTrial.gov before submission of the first version of the manuscript. We added dedicated sentence in the section “Ethical Considerations and declarations”. Lines: 348-349.

“This research was co-funded by the Italian Complementary National Plan PNC-I.1 "Research initiatives for innovative technologies and pathways in the health and welfare sector” D.D. 931 of 06/06/2022, "DARE - DigitAl lifelong pRevEntion" initiative, code PNC0000002.”

We thank for the comment. We added dedicated sentence in the section “Funding”. Lines: 65-66.

We thank for the comment. The minimal anonymized data set necessary to replicate study findings is openly available from BioStudies, S-BSST1682 (https://www.ebi.ac.uk/biostudies/studies/S-BSST1682. DOI: 10.6019/S-BSST1682). Lines: 72-73.

5. Please include a caption for figures 2 and 3.

We thank for the comment. We left the first figure as a graphical abstract, if possible, according to editorial guidelines, and added captions for the other 2 figures. Lines: 412-420.

We thank for the comment. The manuscript was modified accordingly. Lines: 409-420.

Reviewers' comments:

Reviewer #1: The protocol titled "Protocol the Evolution of CROSSWORD Study: Implementing an interoperable web-based platform to support health surveillance against latent tuberculosis infection in healthcare workers and students to define prevention strategies and interventions" is well-structured and addresses a crucial area in tuberculosis (TB) prevention. This study has the potential to make a significant impact on the development of TB preventive strategies in low-TB-burden countries, especially for healthcare workers (HCWs) and students. However, some aspects need clarification and further elaboration:

1. Methods for LTBI Detection Tests:

The manuscript did not clearly define the methods for latent tuberculosis infection (LTBI) detection tests. Both Interferon-Gamma Release Assay (IGRA) and Tuberculin Skin Test (TST) were mentioned in determining the primary endpoint. It is essential to specify whether the IGRA or TST will be used. Since the study involves the conversion of TST, multiple tests are assumed to be done. Information on how often follow-up testing will occur is needed to provide a comprehensive understanding of the testing process.

We thank the Reviewer for the comment. The diagnosis of LTBI requires both conversion to tuberculin after a documented negative tuberculin skin test (TST) at baseline and subsequent positive QuantiFERON-TB Gold test result performed on the same day as the positive TST result. The manuscript was modified accordingly. Lines: 256-262.

2. Interaction with the Developed Web-Based Framework:

The article mentions the development of a web-based interactive platform but does not elaborate on how this framework will engage in the prediction and detection of LTBI. It would be helpful to describe how the platform will interact with collected data, monitor participants, or offer decision support for LTBI prediction.

We thank the Reviewer for the comment. We modified the section “Statistical Analysis” describing the enabling technologies that the research team is adopting for the purposes of the project and mentioning the web-based platform that will be developed as final deliverable of the project. Lines: 332-337.

3. Rationale for the Estimated Prevalence:

The protocol estimates an LTBI prevalence of 1% in the participating institutes. However, the rationale behind this estimate is not sufficiently explained. Is it derived from the ongoing CROSSWORD study? Clarifying how this estimate was derived, considering factors such as the previous prevalence rates in similar settings or literature, would strengthen the study's methodology.

We thank the Reviewer for the comment. The assumed prevalence of LTBI of 1% was based on both the prevalence rate of LTBI (0.62%) among healthcare students and postgraduates described by Verso et al. [Ref. 26] and a reasonable estimate derived from the LTBI prevalence (2.1%) described by Coppeta et al. [Ref. 29] in a population of Italian HCWs in Rome. The manuscript was modified accordingly. Lines: 222-226.

4. Consideration of Changes in Participants’ Roles Over Time:

Since the study involves a potentially prolonged follow-up period, there is a high likelihood that participants’ roles and exposure risks may change (e.g., from students to practicing HCWs). This factor should be taken into account when identifying risk factors and analysing LTBI risk. Clarifying how these changes will be tracked and integrated into the analysis is crucial for ensuring accurate risk assessment.

We thank the Reviewer for the comment. The section “Statistical Analysis” was modified accordingly. Lines: 328-330.

Overall Recommendations:

The study protocol is valuable and has the potential to contribute significantly to TB prevention efforts. Addressing these points will enhance the clarity and comprehensiveness of the study. The authors are encouraged to provide detailed explanations regarding the LTBI detection protocols, web-based framework interaction, rationale for prevalence estimation, and adjustments for role changes during the study follow-up.

We thank the Reviewer for the comment. The manuscript was modified accordingly.

Reviewer #2: The authors describe the protocol of a longitudinal study (retrospective and prospective), aimed at providing a large amount of data on the incidence and prevalence of latent tuberculosis infection, on the incidence of tuberculosis and on the results of screening tests among health workers and students enrolled in 3 university hospitals in Central and Southern Italy. The objective would be achieved through the digital registration of some selected variables reported in a self-administered questionnaire. The objective is ambitious, and the study protocol could validly contribute to scientific progress on a pathology present throughout the world, slightly increasing after the pandemic and with a significant socioeconomic burden even in low-incidence countries, such as Italy. Moreover, university students, an understudied population strata, are included.

However, I think that the submitted description of the protocol should be enriched with details and few methodological aspects should be further clarify. I recommend substantial revisions that take into account the considerations reported below.

We thank the Reviewer for the comment.

First of all, it's necessary to know how many employees and students are in the 3 hospitals considered, this to have information on the feasibility of reaching the sample size in the expected period. It might be useful to know how many people are divided into different age groups, how many students, whether only medical courses or other types of courses, such as engineering, are included, how many men and how many women. It might also be useful to know what is the incidence of tuberculosis in the reference population of the 3 hospitals considered and what is the estimated incidence of active or latent tuberculosis in previous periods among the healthcare workers and students. As regards the feasibility, add information on the progress of the Crossword study, which represents the retrospective phase of the study protocol. Moreover, it's necessary to provide more details on which aspects of the Periskope-TB study were captured in the described protocol, to better support the choice of selected variables.

We thank the Reviewer for the comment. We specified the potential involvement of all employees and students attending medical, specialization and health professions degree courses in the 3 university hospital centers involved in the project making it possible to reach the target population. We modified the section “The sample size”. Lines: 222-240.

Regarding the incidence of tuberculosis in the reference population of the 3 hospitals considered, we have published data on prevalence rate of LTBI (0.62%) among healthcare students and postgraduates described by Verso et al. [Ref. 26]. Therefore, we assumed a reasonable estimate derived also from the LTBI prevalence (2.1%) described by Coppeta et al. [Ref. 29] in a population of Italian HCWs in Rome. The manuscript was modified accordingly. Lines: 222-225.

Regarding the retrospective phase of the study, the final data analysis will be completed by November 2024. We added this detail in the manuscript. Lines: 360-361.

Finally, we specified the predictors, reported by Gupta et al. [Ref. 4] in the final version of prognostic model described in PERISKOPE-TB, included in our questionnaire. We modified the manuscript accordingly. Lines: 179-181.

The DARE project is mentioned but its contribution to this protocol is unclear. More information on the Dare study (bibliographic citation, if available) could clarify how the predictive algorithms adopted in this protocol were digitized and created.

We thank the Reviewer for the comment. We added details on DARE Project and reference n° 28. Lines: 200-205.

It's necessary to stratify the sample size calculation by workers and students, sex and age groups, considering that the enrolled population includes workers and students probably in a wide age range.

Since workers and students are subjected to annual health surveillance, it is not clear whether they will answer only once or multiple times (and how many?) to the proposed questionnaire.

We thank the Reviewer for the comment. We reviewed and reported the formula adopted for the sample size based on both the first and the second outcomes. The stratification of sample size calculation in subgroups was not a priori planned due to expected impact on the number of subjects to be enrolled, mining the feasibility of the whole project. Lines: 222-240.

HCWs and students will respond to the questionnaire survey only once. We modified the manuscript accordingly. Lines: 289-290.

Authors should consider adding to the exclusion criteria, having had tuberculosis or having already undergone chemoprophylaxis.

We thank the Reviewer for the comment. We modified the manuscript accordingly. Lines: 248-249.

About the primary outcomes, consider adding a symptom questionnaire. It should also be clarified how to consider the possible movement between working units. Among the secondary outcomes, it would be better to add a definition of exposure to TB-infected patients, to avoid different descriptions.

We thank the Reviewer for the comment, but symptom assessment was not included in the study protocol. The possible movement between working units is considered among the list of possible risk factors. The meaning of exposure to an infected patient was clarified. We modified the manuscript accordingly. Lines: 275-276.

I would like the auth

---

## [Decision Letter · Decision Letter 1]

5 Jan 2025

PONE-D-24-34897R1Protocol the Evolution of CROSSWORD Study: Implementing an interoperable web-based platform to support health surveillance against latent tuberculosis infection in health care workers and students to define prevention strategies and interventionsPLOS ONE

Dear Dr. Inchingolo,

Thank you for submitting your manuscript to PLOS ONE. After careful consideration, we feel that it has merit but does not fully meet PLOS ONE’s publication criteria as it currently stands. Therefore, we invite you to submit a revised version of the manuscript that addresses the points raised during the review process.

Dear Authors, your manuscript has been greatly improved and both reviewers agree on minor revisions. Please submit the revision by the deadline and also provide a general check of the English language. Best regards==============================

We look forward to receiving your revised manuscript.

Kind regards,

Nicola Serra

Academic Editor

PLOS ONE

Journal Requirements:

Reviewers' comments:

Reviewer's Responses to Questions

**Comments to the Author**

1. Does the manuscript provide a valid rationale for the proposed study, with clearly identified and justified research questions?

Reviewer #1: Yes

Reviewer #3: Yes

2. Is the protocol technically sound and planned in a manner that will lead to a meaningful outcome and allow testing the stated hypotheses?

Reviewer #1: Yes

Reviewer #3: Yes

3. Is the methodology feasible and described in sufficient detail to allow the work to be replicable?

Reviewer #1: No

Reviewer #3: Yes

4. Have the authors described where all data underlying the findings will be made available when the study is complete?

Reviewer #1: Yes

Reviewer #3: Yes

5. Is the manuscript presented in an intelligible fashion and written in standard English?

Reviewer #1: Yes

Reviewer #3: Yes

6. Review Comments to the Author

You may also provide optional suggestions and comments to authors that they might find helpful in planning their study.

Reviewer #1: 1. The author described the primary outcome as prevalence of LTBI. The LTBI was defined as positive TST plus positive QuantiFERON for participants with a baseline negative TST. Will participants with a baseline positive TST be excluded from the cohort? Base one such definition, the primary outcome seems to be the incidence of TB infection instead of prevalence of TB infection.

2. How many TST tests will be done for each participant? When will a follow-up TST test be done for a participant?

Reviewer #3: Management of latent tuberculosis infection (LTBI) is a key component of any TB elimination strategy, as people with LTBI represent a large "human reservoir" for the disease. In these purposes the European Centre for Disease Prevention and Control (ECDC), provides evidence-based guidance for the implementation of public health programmes for the management of LTBI in the European Union (EU) and the European Economic Area (EEA), which includes, beyond the prevalence of the LBT, the identification of risk groups. In this context, the present article fits well.

The authors made most of the changes requested by the reviewer 2; in the references the 7 must be corrected.

In addition, if possible, choose for pubblication a title more impactful, which is currently too long.

7. PLOS authors have the option to publish the peer review history of their article (what does this mean? ). If published, this will include your full peer review and any attached files.

**Do you want your identity to be public for this peer review?** For information about this choice, including consent withdrawal, please see our Privacy Policy .

Reviewer #1: No

Reviewer #3: No

---

## [Author Response · Author response to Decision Letter 1]

15 Jan 2025

January, 15th 2025

To Editor and Reviewers

PLOS ONE

We would like to greatly thank the Editor and Reviewers who encouraged a further revision of the manuscript.

Please find enclosed the revised version (vers. 3) of the Study Protocol article entitled “Protocol the Evolution of CROSSWORD Study: Implementing an interoperable web-based platform to support health surveillance against latent tuberculosis infection in health care workers and students to define prevention strategies and interventions” by Angela Rizzi, Eleonora Nucera, Walter Mazzucco, Pierpaolo Palumbo, Domenico Staiti, Umberto Moscato, Francesco Maria De Simone, Michela Sali, Luca Boldrini, Nikola Dino Capocchiano, Stefano Patarnello, Gabriele Rumi, Raffaella Chini, Valentina Carusi, Michele Centrone, Alessia Di Rienzo, David Longhino, Chiara Laface, Sabato Mellone, Carmelo Massimo Maida, Emanuele Cannizzaro, Luigi Cirrincione, Maria Gabriella Verso, Annalisa Saracino, Francesco Di Gennaro, Luigi Vimercati, Luigi De Maria, Stefania Sponselli, Giancarlo Scoppettuolo, Roberta Pastorino, Antonio Gasbarrini and Riccardo Inchingolo.

PONE-D-24-34897

Protocol the Evolution of CROSSWORD Study: Implementing an interoperable web-based platform to support health surveillance against latent tuberculosis infection in health care workers and students to define prevention strategies and interventions

PLOS ONE

ACADEMIC EDITOR:

Dear Authors,

your manuscript has been greatly improved, and both reviewers agree on minor revisions. Please submit the revision by the deadline and provide a general check of the English language.

We thank the Academic Editor for the comment. We modified the manuscript accordingly.

Reviewers' comments:

Reviewer #1: The author described the primary outcome as prevalence of LTBI. The LTBI was defined as positive TST plus positive QuantiFERON for participants with a baseline negative TST. Will participants with a baseline positive TST be excluded from the cohort? Base one such definition, the primary outcome seems to be the incidence of TB infection instead of prevalence of TB infection.

2. How many TST tests will be done for each participant? When will a follow-up TST test be done for a participant?

We thank the Reviewer for the comment. We detailed the definition of the primary outcome including both the evidence of positive results of TST at enrollment and tuberculin conversion defined as negative TST before enrollment and positive at enrollment. All criteria are verified at enrollment due to cross-sectional nature of the study protocol. Furthermore, we added the exclusion criterion of positive TST performed within 12 months prior to enrollment to exclude subjects from the cohort and specified the number of scheduled TST for each subject. Finally, we specified that the study protocol does not provide follow-up for research purposes, however, each participant continues regular annual follow-up as per routine clinical practice. Lines: 249-265.

Reviewer #3: Management of latent tuberculosis infection (LTBI) is a key component of any TB elimination strategy, as people with LTBI represent a large "human reservoir" for the disease. In these purposes the European Centre for Disease Prevention and Control (ECDC), provides evidence-based guidance for the implementation of public health programmes for the management of LTBI in the European Union (EU) and the European Economic Area (EEA), which includes, beyond the prevalence of the LBT, the identification of risk groups. In this context, the present article fits well.

The authors made most of the changes requested by the reviewer 2; in the references the 7 must be corrected. In addition, if possible, choose for publication a title more impactful, which is currently too long.

We thank the Reviewer for the comment. We removed reference n°7 from the list due to inability to access. Finally, we modified the title as required. Lines: 4-6.

With the best regards,

Angela Rizzi, Eleonora Nucera, Walter Mazzucco, Pierpaolo Palumbo, Domenico Staiti, Umberto Moscato, Francesco Maria De Simone, Michela Sali, Luca Boldrini, Nikola Dino Capocchiano, Stefano Patarnello, Gabriele Rumi, Raffaella Chini, Valentina Carusi, Michele Centrone, Alessia Di Rienzo, David Longhino, Chiara Laface, Sabato Mellone, Carmelo Massimo Maida, Emanuele Cannizzaro, Luigi Cirrincione, Maria Gabriella Verso, Annalisa Saracino, Francesco Di Gennaro, Luigi Vimercati, Luigi De Maria, Stefania Sponselli, Giancarlo Scoppettuolo, Roberta Pastorino, Antonio Gasbarrini and Riccardo Inchingolo.

Corresponding Author:

Riccardo Inchingolo, MD, PhD

UOC Pneumologia, Fondazione Policlinico Universitario A. Gemelli IRCCS. Largo A. Gemelli, 8 – 00168 – Rome, Italy.

riccardo.inchingolo@policlinicogemelli.it

Corresponding Author will receive all editorial communications

The authors declare that the manuscript, or specified parts of it, have not been and will not be submitted elsewhere for publication.

---

## [Decision Letter · Decision Letter 2]

28 Jan 2025

PONE-D-24-34897R2An interoperable web-based platform to support health surveillance against latent tuberculosis infection in health care workers and students: the Evolution of CROSSWORD Study ProtocolPLOS ONE

Dear Dr. Inchingolo,

Thank you for submitting your manuscript to PLOS ONE. After careful consideration, we feel that it has merit but does not fully meet PLOS ONE’s publication criteria as it currently stands. Therefore, we invite you to submit a revised version of the manuscript that addresses the points raised during the review process.

**ACADEMIC EDITOR: ** Dear Authors,Please consider an important final suggestion from Reviewer 1 regarding the timing of the QuantiFERON-TB Gold (QFT) test. Best regards, Nicola Serra

We look forward to receiving your revised manuscript.

Kind regards,

Nicola Serra

Academic Editor

PLOS ONE

Journal Requirements:

Reviewers' comments:

Reviewer's Responses to Questions

**Comments to the Author**

1. Does the manuscript provide a valid rationale for the proposed study, with clearly identified and justified research questions?

Reviewer #1: Yes

Reviewer #3: Yes

2. Is the protocol technically sound and planned in a manner that will lead to a meaningful outcome and allow testing the stated hypotheses?

Reviewer #1: Yes

Reviewer #3: Partly

3. Is the methodology feasible and described in sufficient detail to allow the work to be replicable?

Reviewer #1: Yes

Reviewer #3: Yes

4. Have the authors described where all data underlying the findings will be made available when the study is complete?

Reviewer #1: Yes

Reviewer #3: Yes

5. Is the manuscript presented in an intelligible fashion and written in standard English?

Reviewer #1: Yes

Reviewer #3: Yes

6. Review Comments to the Author

You may also provide optional suggestions and comments to authors that they might find helpful in planning their study.

Reviewer #1: The author addressed my concern. One minor suggestion:

I appreciate the comprehensive definition of LTBI used in the study; however, I would like to suggest a minor revision regarding the timing of the QuantiFERON-TB Gold (QFT) test. The current definition requires the QFT test to be performed on the same day as the positive TST result. In clinical practice, logistical challenges such as laboratory processing delays, sample collection issues, or the need for repeat blood draws may prevent same-day testing. This strict requirement could lead to unnecessary exclusions due to protocol deviations, even in cases where the QFT result is positive within a reasonable timeframe. To improve the feasibility of the study without compromising diagnostic accuracy, I recommend allowing a reasonable time window for QFT testing rather than requiring it strictly on the same day. This adjustment would better align with real-world clinical workflows and help avoid potential difficulties in data interpretation and patient classification.

Reviewer #3: The authors made the changes required and now I belive that the articol can be accepted.

7. PLOS authors have the option to publish the peer review history of their article (what does this mean? ). If published, this will include your full peer review and any attached files.

**Do you want your identity to be public for this peer review?** For information about this choice, including consent withdrawal, please see our Privacy Policy .

Reviewer #1: No

Reviewer #3: No

---

## [Author Response · Author response to Decision Letter 2]

1 Feb 2025

February, 1st 2025

To Editor and Reviewers

PLOS ONE

We would like to greatly thank the Editor and Reviewers who encouraged a further revision of the manuscript.

Please find enclosed the revised version (vers. 4) of the Study Protocol article entitled “An interoperable web-based platform to support health surveillance against latent tuberculosis infection in health care workers and students: the Evolution of CROSSWORD Study Protocol” by Angela Rizzi, Eleonora Nucera, Walter Mazzucco, Pierpaolo Palumbo, Domenico Staiti, Umberto Moscato, Francesco Maria De Simone, Michela Sali, Luca Boldrini, Nikola Dino Capocchiano, Stefano Patarnello, Gabriele Rumi, Raffaella Chini, Valentina Carusi, Michele Centrone, Alessia Di Rienzo, David Longhino, Chiara Laface, Sabato Mellone, Carmelo Massimo Maida, Emanuele Cannizzaro, Luigi Cirrincione, Maria Gabriella Verso, Annalisa Saracino, Francesco Di Gennaro, Luigi Vimercati, Luigi De Maria, Stefania Sponselli, Giancarlo Scoppettuolo, Roberta Pastorino, Antonio Gasbarrini and Riccardo Inchingolo.

PONE-D-24-34897R2

An interoperable web-based platform to support health surveillance against latent tuberculosis infection in health care workers and students: the Evolution of CROSSWORD Study Protocol

PLOS ONE

ACADEMIC EDITOR:

Dear Authors,

Please consider an important final suggestion from Reviewer 1 regarding the timing of the QuantiFERON-TB Gold (QFT) test.

We thank the Academic Editor for the comment. We modified the manuscript accordingly.

Reviewers' comments:

Reviewer #1: The author addressed my concern. One minor suggestion:

I appreciate the comprehensive definition of LTBI used in the study; however, I would like to suggest a minor revision regarding the timing of the QuantiFERON-TB Gold (QFT) test. The current definition requires the QFT test to be performed on the same day as the positive TST result. In clinical practice, logistical challenges such as laboratory processing delays, sample collection issues, or the need for repeat blood draws may prevent same-day testing. This strict requirement could lead to unnecessary exclusions due to protocol deviations, even in cases where the QFT result is positive within a reasonable timeframe. To improve the feasibility of the study without compromising diagnostic accuracy, I recommend allowing a reasonable time window for QFT testing rather than requiring it strictly on the same day. This adjustment would better align with real-world clinical workflows and help avoid potential difficulties in data interpretation and patient classification.

We thank the Reviewer for the comment. To improve the feasibility of the study without compromising diagnostic accuracy, we introduced a reasonable time window of 5days for QFT testing. We modified accordingly. Lines: 257-258.

Reviewer #3: The authors made the changes required and now I believe that the article can be accepted.

We thank the Reviewer for the comment.

With the best regards,

Angela Rizzi, Eleonora Nucera, Walter Mazzucco, Pierpaolo Palumbo, Domenico Staiti, Umberto Moscato, Francesco Maria De Simone, Michela Sali, Luca Boldrini, Nikola Dino Capocchiano, Stefano Patarnello, Gabriele Rumi, Raffaella Chini, Valentina Carusi, Michele Centrone, Alessia Di Rienzo, David Longhino, Chiara Laface, Sabato Mellone, Carmelo Massimo Maida, Emanuele Cannizzaro, Luigi Cirrincione, Maria Gabriella Verso, Annalisa Saracino, Francesco Di Gennaro, Luigi Vimercati, Luigi De Maria, Stefania Sponselli, Giancarlo Scoppettuolo, Roberta Pastorino, Antonio Gasbarrini and Riccardo Inchingolo.

Corresponding Author:

Riccardo Inchingolo, MD, PhD

UOC Pneumologia, Fondazione Policlinico Universitario A. Gemelli IRCCS. Largo A. Gemelli, 8 – 00168 – Rome, Italy.

riccardo.inchingolo@policlinicogemelli.it

Corresponding Author will receive all editorial communications

The authors declare that the manuscript, or specified parts of it, have not been and will not be submitted elsewhere for publication.

---

## [Decision Letter · Decision Letter 3]

5 Feb 2025

An interoperable web-based platform to support health surveillance against latent tuberculosis infection in health care workers and students: the Evolution of CROSSWORD Study Protocol

PONE-D-24-34897R3

Dear Dr. Inchingolo,

We’re pleased to inform you that your manuscript has been judged scientifically suitable for publication and will be formally accepted for publication once it meets all outstanding technical requirements.

Kind regards,

Nicola Serra

Academic Editor

PLOS ONE

Additional Editor Comments (optional):

Dear Authors,

Congratulations for the great work done. Your article is now ready for publication

Best regards.

Nicola Serra

Reviewers' comments:

Reviewer's Responses to Questions

**Comments to the Author**

1. Does the manuscript provide a valid rationale for the proposed study, with clearly identified and justified research questions?

Reviewer #1: Yes

2. Is the protocol technically sound and planned in a manner that will lead to a meaningful outcome and allow testing the stated hypotheses?

Reviewer #1: Yes

3. Is the methodology feasible and described in sufficient detail to allow the work to be replicable?

Reviewer #1: Yes

4. Have the authors described where all data underlying the findings will be made available when the study is complete?

Reviewer #1: Yes

5. Is the manuscript presented in an intelligible fashion and written in standard English?

Reviewer #1: Yes

6. Review Comments to the Author

You may also provide optional suggestions and comments to authors that they might find helpful in planning their study.

Reviewer #1: Thanks to the authors for the revision. I have no further comments. I recommend accept the paper as current version.

7. PLOS authors have the option to publish the peer review history of their article (what does this mean? ). If published, this will include your full peer review and any attached files.

**Do you want your identity to be public for this peer review?** For information about this choice, including consent withdrawal, please see our Privacy Policy .

Reviewer #1: No

---

## [Editor Report · Acceptance letter]

PONE-D-24-34897R3

PLOS ONE

Dear Dr. Inchingolo,

I'm pleased to inform you that your manuscript has been deemed suitable for publication in PLOS ONE. Congratulations! Your manuscript is now being handed over to our production team.

Kind regards,

on behalf of

Dr. Nicola Serra

Academic Editor

PLOS ONE